# The Interaction of Gut Microbiota and Heart Failure with Preserved Ejection Fraction: From Mechanism to Potential Therapies

**DOI:** 10.3390/biomedicines11020442

**Published:** 2023-02-02

**Authors:** Wei Yu, Yufeng Jiang, Hui Xu, Yafeng Zhou

**Affiliations:** 1Department of Cardiology, Dushu Lake Hospital Affiliated to Soochow University, Medical Center of Soochow University, Suzhou Dushu Lake Hospital, Suzhou 215000, China; 2Institute for Hypertension, Soochow University, Suzhou 215000, China

**Keywords:** gut microbiota, heart failure with preserved ejection fraction, trimethylamine N-oxide, short-chain fatty acids, bile acids, inflammation, metabolic syndrome, therapies

## Abstract

Heart failure with preserved ejection fraction (HFpEF) is a disease for which there is no definite and effective treatment, and the number of patients is more than 50% of heart failure (HF) patients. Gut microbiota (GMB) is a general term for a group of microbiota living in humans’ intestinal tracts, which has been proved to be related to cardiovascular diseases, including HFpEF. In HFpEF patients, the composition of GMB is significantly changed, and there has been a tendency toward dysbacteriosis. Metabolites of GMB, such as trimethylamine N-oxide (TMAO), short-chain fatty acids (SCFAs) and bile acids (BAs) mediate various pathophysiological mechanisms of HFpEF. GMB is a crucial influential factor in inflammation, which is considered to be one of the main causes of HFpEF. The role of GMB in its important comorbidity—metabolic syndrome—also mediates HFpEF. Moreover, HF would aggravate intestinal barrier impairment and microbial translocation, further promoting the disease progression. In view of these mechanisms, drugs targeting GMB may be one of the effective ways to treat HFpEF. This review focuses on the interaction of GMB and HFpEF and analyzes potential therapies.

## 1. Introduction

Heart failure (HF) is a group of clinical syndromes caused by pathological changes of cardiac structure and (or) function, often manifested as fatigue, edema (including systemic congestion, pulmonary congestion, third space effusion), and dyspnea, etc. According to the left ventricular ejection fraction (LVEF), HF is divided into three types: HF with reduced ejection fraction (HFrEF) (LVEF < 40%), HF with mildly reduced ejection fraction (HFmrEF) (40% ≤ LVEF < 50%), and HF with preserved ejection fraction (HFpEF) (LVEF ≥ 50%). HFpEF is highly prevalent, accounting for 50% or more of all HF patients and is associated with significant mortality, mainly chronic morbidity and high frequency of re-hospitalizations [1,2,3]. However, there is no clear pathogenesis or accurate and effective treatment for HFpEF, which means its prevalence will further increase and it will become a more serious public health problem.

There are a large number of microbes living in human intestines, which have a symbiotic relationship with human beings. They can affect human bodies by metabolism, immunity, secretion, or in other ways, while the vivo environment will also affect their growth and function. The interaction between gut microbiota (GMB) and their host has been a research hotspot in recent years. A lot of researchers have demonstrated that the GMB are strongly related to cardiovascular problems, such as atherosclerosis, myocardial infarction, hypertension, arrhythmia, and HF. HF, especially HFrEF, has been extensively studied, but relatively less has been done on HFpEF. This review focused on the possible interaction between GMB and HFpEF and proposed some potential treatments.

## 2. The Features of GMB in HFpEF

Although there are not enough direct studies on the GMB composition of HFpEF patients, most of the available results indicated significant differences in the community structure between HFpEF patients and healthy people, with a gut microbial dysbiosis in HFpEF patients. Huang et al.’s research, including 30 HFpEF patients and 30 healthy controls, showed an increasing abundance of flora that could trigger inflammatory response (Lactobacillus and Enterococcus) paralleled by a decrease of anti-inflammatory flora (Butyricicoccus, Sutterella, Lachnospira, and Ruminiclostridium) [4]. Beale et al. found that the Ruminococcus—a bacterium known to be a producer of short-chain fatty acids (SCFAs)—was depleted and this change was the driving factor for the difference between HFpEF patients and control groups in this research [5]. These results indicated that GMB could affect the heart by producing inflammatory mediators or metabolites. Both of these two studies reported a reduction of the Chao index (reflecting the species richness) in HFpEF patients in the alpha diversity analysis and a significant divergence of the HFpEF and the control group in beta diversity analysis. Lower alpha diversity is often a marker of microbial dysbiosis, and the beta diversity showed a clear inter-group difference in GMB [6].

In addition, an association between the composition of microbiota and some markers of myocardial fibrosis has been reported, such as C-terminal propeptide of procollagen type I (PICP), N-terminal propeptide of pro-collagen type III (PIIINP), and left ventricular extracellular volume (ECV) [7,8], which demonstrated that GMB may affect the myocardial structure. Studies about the GMB in HFpEF are shown in Table 1.

Apparently, existing research on the composition of intestinal flora in HFpEF patients is insufficient. The composition of GMB differs among regions, races and diets. Therefore, larger sample size, multi-region, and multi-center studies on the GMB of HFpEF patients can provide a more reliable theoretical basis for the treatment of microbial adjustment.

## 3. The Mechanisms by Which GMB Interacts with HFpEF

Unfortunately, HFpEF has great pathophysiological heterogeneity among different patients, so there is no accurate and detailed theory that can explain the pathogenesis of all patients. However, some pathophysiological changes have been shown to be associated with the occurrence and development of HFpEF, including left heart diastolic/systindiolic dysfunction, pulmonary vascular disease, right heart dysfunction, inflammation, cellular and extracellular structural changes, cardiometabolic abnormalities, autonomic dysregulation, cardiac fibrosis, vascular stiffening, myocardial ischemia, endothelial dysfunction, etc. Diastolic dysfunction is a basic component of the pathophysiology [10,11]. GMB affects HFpEF by intervening in these processes through metabolism and inflammation. In addition, the dysregulation of GMB, intestinal barrier dysfunction, and microbial translocation derived from cardiac insufficiency will further aggravate HFpEF.

### 3.1. Intestinal Barrier Dysfunction and Microbial Translocation

Intestinal barrier dysfunction is always observed in HF patients [12]. The intestinal barrier is composed of mechanical, chemical, immunological, and biological barriers. Normally, the GMB forms an important protective barrier against pathogens. When the stability of this biocoenosis is disrupted, intestinal colonization resistance is severely impaired, making invasion and colonization by potential pathogens (including conditionally pathogenic bacteria) easier in the gut. Much evidence supported the indication that patients with HFpEF have a gut microbial dysregulation in comparison with healthy people [4,5,9], meaning the risk of infection is increased after the onset of HFpEF. The chronic heart failure (CHF) population has a large number of pathogenic bacteria compared to healthy controls, such as Candida, Campylobacter, Salmonella, Shigella, and Yersinia enterocolitica [13]. Moreover, insufficient cardiac output in heart failure could lead to intestinal ischemia, edema, and inflammation, increasing the permeability of the bowel wall and aggravating the compromised epithelial barrier [14].

Bacterial translocation is defined as the penetration of a large number of tissues and organs outside the intestine by normal microbiota and their endotoxins, peptidoglycans, or metabolites that initially colonized in the intestine through the intestinal mucosal barrier [15]. It is often secondary to intestinal barrier dysfunction. Zhou et al. found that the blood flora composition of patients with cardiac dysfunction after acute myocardial infarction has apparently changed compared with healthy people or patients with stable coronary heart disease [16]. Plasma endotoxin and cytokine levels are also raised in CHF patients [17,18]. The levels of endotoxin in the hepatic veins was higher than in the left ventricle during acute HF, which directly suggested that the bacterium or endotoxin did translocate from intestine into bloodstream [19]. Endotoxin is one of the most important pro-inflammatory activators, and inflammation is a crucial part of the pathogenesis of HFpEF, so endotoxin could play an important role in this disease.

### 3.2. Inflammation

A lot of experimental and clinical evidence has demonstrated the critical role of inflammation in the occurrence and development of HEpEF by direct injury or from interacting with other pathophysiological changes. Regardless of the primary etiology, HF is related to both local and systemic inflammatory responses [20]. HFpEF patients showed higher levels of inflammation than HFrEF patients, and HFpEF is more affected by systemic inflammation, while the heart insult of HFrEF is mainly based on a local inflammatory reaction [21,22]. A subgroup of inflammatory cells expressing profibrotic growth factor transforming growth factor-β (TGF-β) can be found in endomyocardial biopsy samples from HFpEF patients [23]. In patients with HFpEF, the increase in plasma content of inflammatory indicators, including soluble interleukin (IL)-1 receptor-like 1, growth differentiation factor 15, soluble ST2, and pentraxin-3, is greater than in HFrEF or other conditions [24]. Chirinos et al. screened TNF-alpha, sTNFRI, and IL-6 as predictors of all-cause death or heart failure-related hospital admission risk by machine-learning [25].

High levels of inflammatory factors could lead to endothelial dysfunction, myocardial fibrosis or other pathological changes that then promote the progress of HFpEF [26,27,28,29]. For example, inflammatory mediators (inflammatory cytokines, chemokines, and growth factors) are related to the pathogenesis of cardiac fibrosis by directly activating fibroblasts, stimulating the recruitment and activation of fibrotic macrophages and lymphocytes, and triggering the fibrotic process in vascular cells and cardiac myocytes. Long-term chronic inflammation may lead to myocardial cell necrosis and cause fibrosis in the form of repair [30].

In addition, recent studies reported that the NLR family pyrin domain containing 3 (NLRP3) inflammasome may be critical for the systemic inflammatory response and cardiac remodeling of HFpEF. Activation of the NLRP3 inflammasome and secondary IL-1β/IL-18 overproduction were found in HFpEF mouse models [31,32]. NLRP3 inflammasome in endothelial cells will disrupt the tight connection between cells, resulting in endothelial dysfunction [33]. The NLRP3 inflammasome may also induce ventricular arrhythmias in HFpEF patients, leading to poor prognosis [34]. Inhibition of the NLRP3 inflammasome reduces inflammation and fibrosis and reverses cardiac remodeling [35].

GMB is associated with systemic inflammation levels. There is already evidence of an increase in the abundance of pro-inflammatory bacteria and a decline in anti-inflammatory bacteria in patients with HFpEF [4]. In addition, increased bacterial diversity is always accompanied by decreased levels of inflammatory markers [36], because the dysbacteriosis may break immune homeostasis, leading to the occurrence of inflammation. Both the structural components and metabolites of bacteria can enter the bloodstream through microbial translocation or be delivered to immune cells and trigger an inflammatory cascade [37]. Endotoxin is a typical pro-inflammatory component of bacteria, and pathophysiology-related endotoxin amounts have been shown to induce the release of multiple inflammatory factors in vitro or vivo of patients with HF [38,39]. It is mainly combined with Toll-Like receptor 4 (TLR4) on the heart to play an inflammatory role or impair the heart [40], and TLR4 expression was increased on the heart of patients with advanced HF [41]. Blocking its effect with the inhibition of TLR4 may benefit HFpEF patients. Moreover, GMB and its metabolites have regulatory effects on the NLRP3 inflammasome, and dysbacteriosis will cause activation of the NLRP3 inflammasome, which will further aggravate inflammation or myocardial fibrosis [42,43].

In conclusion, when intestinal immune homeostasis is disrupted, the inflammatory cascade triggered by GMB, especially its endotoxins, is a complex immune response system composed of intricate cytokine networks and has not been completely studied. If future studies could screen out the immune factors that play the key role in this network and carry out precise immune targeted therapy on them, it will have important therapeutic significance for systemic inflammation and HFpEF. Of course, targeting microbiota and endotoxins in the upstream of the cascade or NLRP3 is a relatively simple approach, so the effect of TLR4 inhibitors or NLRP3 inhibitors on HFpEF is worth studying.

### 3.3. Metabolites

#### 3.3.1. Trimethylamine N-oxide (TMAO)

TMAO is an amine oxide, which naturally presents in our diets, such as in fish, or that can be generated from its metabolic precursors—food-derived choline, carnitine and other substances. The precursors must first be converted into trimethylamine (TMA) by TMA lyase derived from GMB, then TMA will enter the portal circulation and be oxidized to TMAO by hepatic enzyme flavin-containing monooxygenase 3 in the liver. Diet and endogenous TMAO can be released by the liver and absorbed by extrahepatic tissues or excreted with urine [44,45]. Firmicutes, Proteus, and Actinomycetes have been proven to be involved in this process [46]. In 2011, Wang et al. screened TMAO from more than 2000 small molecule metabolites and demonstrated that increased plasma levels of TMAO could predict the risk of several cardiovascular diseases in a large independent clinical cohort [47].

Patients with HFpEF have more TMAO in the plasma than healthy people [48]. This is not merely an independent hazard factor of HFpEF, but it is also associated with other risk factors such as blood urea nitrogen, creatinine, and N-terminal pro-B-type natriuretic peptide (NT-proBNP) [49]. It is a prognostic predictor of HFpEF. Elevated TMAO was independently associated with an increase in the composite endpoints of HF readmissions and cardiac death in HFpEF patients and this effect was enhanced in patients with malnutrition status [50]. Moreover, it could be used as a marker of risk stratification for HFpEF; especially when BNP is not high, it could more sensitively reflect the prognosis [51].

TMAO may lead to cardiac diastolic dysfunction, which is the key hemodynamic change of HFpEF, and myocardial fibrosis is a significant reason for diastolic dysfunction. A prospective cohort study of 112 samples presented a positive correlation between TMAO and diastolic dysfunction indices, such as mitral E/septal Ea and LA volume index [52]. Chen et al. found an increased level of TMAO derived from a diet high in sugar and saturated fat contributed to inflammation and fibrosis of the heart and impaired diastolic function [53]. TMAO promotes TGF-β/SMAD3 expression. TGF-β activates fibroblasts through the second messenger SMAD and induces the secretion of type I collagen, which is the molecular basis of myocardial and pulmonary fibrosis [54]. Its effects could be blocked by SMAD3 inhibitor or TMA synthesis inhibitor [55]. It has also been demonstrated through in vitro and in vivo experiments that TMAO may aggravate pulmonary hypertension [56]. Furthermore, TMAO increased the expression of pro-inflammatory genes via the nuclear factor kappa-B (NF-κB) pathway and activated the NLRP3 inflammasome [57]. In short, TMAO derived from GMB intervenes in the development of HFpEF by inducing cardiac remodeling through promoting myocardial fibrosis and pro-inflammatory effects. Blocking its molecular pathway can be used as a drug therapy idea.

#### 3.3.2. SCFAs

SCFAs are organic fatty acids containing less than six carbon atoms; acetic acid, propionic acid and butyric acid are more common in human bodies. SCFAs are products of the fermentation of various fiber substances in food by the GMB, after which they are taken by clonocytes or enter the portal vein and provide energy to the liver cells; a minor fraction of colon-derived SCFAs enters the systemic circulation and is taken advantage of by other tissues [58,59].

Most evidence suggests that SCFAs have a protective effect against HF. In the normal adult heart, energy is mainly derived from the oxidation of long-chain fatty acids (LCFAs). In failing hearts, however, LCFAs produce less ATP, while SCFAs normalize contractile function as an effective energy source [60,61]. Additionally, SCFAs play their physiological roles as antihypertension agents, for example, and by providing protective effects on endothelial function via G protein-coupled receptors (GPCRs) [62]. A high-fiber diet and acetate supplementation significantly reduced blood pressures, cardiac fibrosis, and left ventricular hypertrophy through the downregulation of early growth response 1 (Egr1) [63]. Butyrate alleviated pulmonary hypertension, reversed right ventricular hypertrophy, and preserved diastolic and systolic functions of the heart in rats [64,65]. SCFAs also play an important role in the regulation of immunity and inflammation by two main mechanisms: regulating the expression of inflammatory genes via GPCRs and its downstream NF-κB and MAPK pathways, or by going directly into the cell to inhibit histone deacetylase [66].

SCFAs also have positive effects on intestinal microbial homeostasis and barrier function. Small intestinal bacterial overgrowth (SIBO) is a common manifestation of microbial dysbiosis, which was found in some patients with HFpEF and which increased their risk of cardiovascular death [67]. Butyrate has a significant negative correlation with SIBO [68]. SCFAs could up-regulate tight junction protein claudin-1 transcription to enhance the intestinal epithelial barrier function [69]. In HFpEF patients, SCFAs producing microbiota is reduced [5], which weakens the protective effect of SCFAs on the gut and heart, and further aggravates the disease. Therefore, SCFAs not only improve cardiac function by ensuring the energy supply of the heart, reducing blood pressure, preventing ventricular hypertrophy, and controlling inflammation, but also stabilize the intestinal microbial environment and prevent further damage caused by GMB disorder in HFpEF.

#### 3.3.3. Bile Acids (BAs)

BAs can be classified into primary bile acids (PBAs) and secondary bile acids (SBAs), according to their sources. PBAs are synthesized in the liver by cholesterol, then about 5–10% of them reach the colon and undergo biotransformations by certain microbiota and finally into SBAs such as deoxycholic acid, ursodeoxycholic acid (UDCA), and lithocholic acid (LCA) [70,71]. Moreover, BAs can be divided into hydrophilic and hydrophobic BAs by their molecular structures. Hydrophobic BAs are more toxic to cells due to their high affinity for lipids and have been found to be associated with an increased risk of cardiovascular diseases, while hydrophilic BAs are more beneficial to the heart. LCA has the strongest hydrophobicity, and UDCA has the strongest hydrophilicity [72,73]. UDCA has been proven to have an important cardioprotective effect and to provide a dose-dependent inhibition of myocardial fibrosis [74,75,76].

BAs bind to receptors like farnesoid X-receptor (FXR) and takeda G-protein-coupled receptor 5 (TGR-5), etc., to affect the heart [73]. For example, UDCA, as an agonist of FXR, degraded nitric oxide synthase inhibitors to improve the bioavailability of NO, which mediates calcium desensitization in myofilaments and myocardial relaxation via the PKG pathway, thereby reducing diastolic dysfunction and myocardial fibrosis [75,77]. TGR-5 activates pro-survival kinases and heat shock proteins which could protect cardiac cells in HF and enhances its adaptability to physiological, inotropic and hemodynamic stress in mice [78]. In contrast to TMAOs, BAs inhibit the NLRP3 inflammasome activation via TGR-5 signaling to exert anti-inflammatory effects [79].

BAs also modulate the stability of intestinal ecology. In fact, bile salts have a certain degree of antibacterial activity, which provides a selective pressure on the flora, thus regulating the composition of GMB [80]. PBAs promote the restoration of the microbiome after ecological dysregulation and prevent the overgrowth of pathogenic bacteria in the small intestine [81]. Increased bacteria levels in the ileum and an impaired intestinal barrier were found in mice lacking FXR; thus, FXR could inhibit the bacterial overgrowth and mucosal injury in the ileum [82]. TGR-5 null mice had a disarrayed molecular architecture of colonic tight junctions and a higher intestinal permeability [83].

In conclusion, BAs produce a marked effect, mainly through FXR and TGR-5, to protect cardiomyocytes, improve diastolic function, resist inflammation in HFpEF, and repair intestinal barrier. However, not all BAs have positive effects, such as hydrophobic BAs, so it is necessary to study the impact of different BAs on HFpEF.

#### 3.3.4. Other Metabolites

In addition to TMAOs, SCFAs, and BAs, which have been extensively studied, new potential HFpEF-related metabolites are continually being discovered. For instance, polyphenols are plant-derived antioxidants. Dietary polyphenols are converted into low-molecular-weight, absorbable bioactive metabolites in the small intestine by specific enzymes derived from GMB [84]. In HF mouse models, polyphenols ameliorated HF-induced GMB disorders and protected cardiac function [85,86]. Polyphenols have antioxidant, anti-inflammatory, protective endothelial, and anti-myocardial fibrosis effects, which may be beneficial to HFpEF [84,87,88].

GMB has been shown to be correlated with amino acid (AA) metabolism. Microbes help the host to establish amino acid homeostasis, while AAs regulate the abundance and diversity of AA-fermenting microbiota [89]. The metabolic disorder of AAs is related to the pathophysiological mechanism of HFpEF. For example, HF patients with high plasma levels of phenylalanine have higher levels of C-reactive protein (CRP), inflammatory cytokines (IL-8, IL-10), and higher mortality [90], while glycine shows anti-inflammatory effects and protection of cells and heart [89]. The GMB metabolites of AAs also affect HFpEF. The fermentation product of tryptophan—indoxyl sulphate—increases expression of pro-inflammatory and pro-fibrotic signaling molecules and induces oxidative stress in animal experiments [91,92]. The levels of phenylacetylgutamine—a product of fermentation which has been proven to promote thrombosis—tracks with NT-proBNP levels in HFpEF patients, suggesting the potential correlation between them [92,93].

Timethyl-5-aminovaleric acid (TMAVA) is the metabolite of trimethylysine (TML) through GMB. TML is also the metabolic precursor of TMAO, and TML will be preferentially converted to TMAVA rather than TMAO in mice. Zhao et al. found the relationship between TMAVA and HF in 2022. In a cohort study involving 1647 patients, TMAVA was positively correlated with the risk of HF and the risk of death. In animal experiments, they found that TMAVA could inhibit the synthesis of carnitine, lead to the reduction of fatty acid oxidation, ultimately disrupt the myocardial energy metabolism, and induce the excessive accumulation of myocardial lipids, which is related to the decline of diastolic function [94].

The research on these newly discovered metabolites is still insufficient, and more experimental evidence is needed to verify their association with HFpEF. Of course, searching for new metabolites is an important part of intestinal microbiology and metabolomics. New technologies, like machine learning, can help to screen new metabolic markers.

## 4. Metabolic Syndrome (MetS), HFpEF and GMB

MetS is a series of metabolic disorders including central obesity, insulin resistance, glucose intolerance, hypertension, and dyslipidemia [95]. GMB was found to be involved in MetS. Ridaura et al. transplanted the GMB from one obese twin and one thin twin into germ-free mice. The mice that received the obese twin microbiota significantly tended to become obese [96], proving an essential impact of GMB on obesity. People with low gut microbiota abundance were more likely to have obesity, dyslipidemia, and insulin resistance [97]. SCFAs, Bas, and inflammation also play a complex regulatory role between GMB and MetS [98]. The gut-centric view of MetS has been proposed because of these associations [99].

Meanwhile, MetS is a classical hazard factor of cardiovascular diseases, particularly as an important comorbidity of HFpEF. Among various cardiovascular diseases, obesity has the strongest correlation with HF and more significantly raises incidence rate and mortality of HFpEF compared with HFrEF [100,101]. From an epidemiological point of view, HFpEF patients are mainly elderly people, of which 84% have overweight/obesity, more than 60% have hypertension, and over 20% have type 2 diabetes [102]. In terms of pathophysiology mechanism, insulin resistance can induce endothelial dysfunction and myocardial energy failure, and eventually lead to diastolic dysfunction [103]. Metabolic inflammation and oxidative stress stimulate myocardial fibrosis in obese patients [104]. Thus, MetS is an important bridge for GMB to affect HFpEF. When GMB is disturbed, the direct consequence is more likely to be nutritional metabolism disorders such as fat accumulation and insulin resistance, which increases the probability of HFpEF. When MetS occurs, we could intervene early in the course of the disease by adjusting GMB to achieve the purpose of HFpEF prevention. So, the GMB therapy of MetS is also a research direction with important clinical significance.

The interaction between GMB and HFpEF is shown in Figure 1. 

## 5. Potential Therapies

### 5.1. Dietary Intervention

In the last decade of microbiome research, diet has been identified as a crucial determinant of the structure and function of GMB [105]. The Mediterranean diet (MD) has been shown in many articles to benefit the gut microbial environment. It features a lot of olive oil, legumes, whole grains, fruits and vegetables, moderate amounts of fish, dairy products and red wine, and a small amount of meat products, containing high levels of dietary fiber, polyunsaturated fatty acids, polyphenols, and a small amount of red meat [106]. MD is associated with greater biodiversity, higher abundance of probiotics, increased SCFAs, and reduced TMAO [107,108]. The modified MD score and the risk of HF was negatively correlated in both men and women [109,110]. Higher MD compliance improved the long-term prognosis of HFpEF in a cohort study followed for ten years. This effect was correlated with CRP levels, suggesting that inflammation may be the pathway by which this diet plays its beneficial role [111]. And MD has the potential to become one of the first treatments for MetS [112]. Dietary intervention is a basic, safe and convenient treatment, thus early use of MD in HFpEF patients may slow the disease progression and improve the prognosis.

### 5.2. Antibiotics

Antibiotics, as strong modulators of microbiome, have been tried to regulate gut microbiota and control inflammation in some earlier studies [113]. Some kinds of antibiotics are beneficial to GMB. For example, nitrofurantoin and rifaximin could promote the growth of probiotics in intestines [114]. Vancomycin promoted the accumulation of chenodeoxycholic acid—a kind of PBAs—by its bactericidal effect on the Clostridium species [115]. Moreover, antibiotic treatment alleviated bacterial migration, inflammation and myocardial injury in animal experiment [16]. However, it is difficult for antibiotics to precisely target harmful bacteria only; the composition of normal bacteria is also easily affected by them, and their toxicity and potential drug resistance are also important risk factors for HFpEF patients with systemic low perfusion. Some classes of antibiotics had adverse impact on cardiovascular events and elevated the cardiovascular death frequency in several clinical research studies; the U.S. Food and Drug Administration even alerted the public about the use in some cardiovascular diseases [116,117,118]. Therefore, in the absence of a clear indication of infection, antibiotics are not good options, which is consistent with the current trend of restricting antibiotics use.

### 5.3. Probiotics and Prebiotics

Probiotics are live microorganisms that could regulate the composition of GMB. Oral Akkermansia muciniphila supplementation could improve the MetS [119]. Probiotic Lactobacillus rhamnosus GR-1 contributed to the restoration of cardiac systolic and diastolic function in rats after six weeks of sustained coronary occlusion [120]. Patients with HF who were treated with S. boulardii for three months also showed significant reductions in creatinine and inflammatory markers [121]. Prebiotics are substances that can be selectively utilized by microorganisms in the host body and converted into integrations beneficial to the health of the host. Fructooligosaccharides, galacto-oligosaccharides, inulin, etc. are prebiotics commonly used that have been widely added to dietary supplements. Prebiotic fibro-deficient mice had a higher susceptibility to hypertension along with a significant increase of tau (left ventricular diastolic constant) and myocardial perivascular fibrosis, indicating the early manifestations of HFpEF [122]. Both endotoxins and opportunistic pathogens were reduced in rats with HF treated with prebiotic complexes [123]. In addition, certain prebiotics have been found to reduce adverse reactions to antibiotics during combination therapy, especially ampicillin and inulin [124]. This approach may ameliorate the impact of inevitable antibiotic use on the prognosis of HFpEF patients.

However, the role of probiotics and prebiotics is limited due to the diversity and complexity of GMB, meaning that it can only be used as an adjunctive therapy. In addition, there are researchers pointing out their potential side effects. For instance, strains of probiotics can directly cause bacteremia in ICU patients and inulin induced liver damage in mice with *dysmicrobiota* [125,126]. HFpEF is often accompanied by dysbacteriosis and immunocompromised conditions, so the timing and dose of probiotics or probiotics supplements need to be carefully considered.

### 5.4. Fecal Microbiota Transplantation (FMT)

FMT, a technique for transplanting functional bacteria from the stool of a healthy person into the intestine of a patient, is mainly used to treat Clostridium difficile infection. It could directly reconstruct healthy GMB and repair the intestinal barrier [127]. There are also randomized control trials (RCTs) demonstrating that FMT contributed to weight loss in obese patients and improvement of insulin sensitivity in MetS patients, and no related adverse events were found, so FMT could at least prevent HFpEF by treating MetS [128,129]. However, no direct studies have proven the curative effect of FMT against HFpEF; this is also a meaningful research direction.

### 5.5. Exercise Training

Exercise training is one of the primary and secondary prevention methods for cardiovascular disease. The beneficial effects of exercise on HFpEF are more notable than those on HFrEF [130]. Exercise training reduced left ventricular myocardial stiffness in patients with preclinical HFpEF [131]. Two meta-analyses both reported that exercise contributed to the cardiopulmonary function and quality of life in HFpEF patients, although diastolic function showed no significant change [132,133]. Furthermore, it could improve the status of gut microbes and increase alpha diversity [134]. Its positive effect on metabolic syndrome is beyond doubt [135,136]. However, we generally think that patients with HF should be very careful when exercising to avoid overloading the heart or causing serious consequences. Therefore, exercise training for HFpEF patients can only be carried out when the condition is stable, and the time and intensity of training need further research to determine.

### 5.6. Targeting Metabolites and Inflammation

Compared with the regulation of GBM structure, targeting downstream metabolites or inflammatory cascade may be a more accurate and effective therapeutic idea. TMAO is clearly harmful to HFpEF patients. The non-lethal microbial enzyme inhibitor of choline TMA lyase has been used to attenuate the TMAO production in animal models. For instance, 3,3-dimethyl-1-butanol (DMB) prevented inflammation, cardiac hypertrophy, and fibrosis in HF mice, and mice did not exhibit toxic reactions after the chronic DMB exposure [137,138]. Iodomethylcholine reversed the changes in heart structure as well as function caused by TMAO and passed a series of pharmacological safety tests [139]. Fluoromethylcholine, as the next-generation choline TMA lyase suicide substrate inhibitor, not only blocked choline decomposition more effectively, but was also difficult for the host to absorb, reducing its chance of entering the blood circulation and causing side effects [140]. Their safety and effectiveness can be preliminarily guaranteed as drugs for HFpEF, but the research on humans is incomplete.

Supplemental SCFAs mainly depend on a higher intake of fiber from diet or prebiotics, or it can also be administered exogenously [141]. Apart from providing energy to a failing heart and anti-inflammation effects, SCFAs can improve metabolic syndrome, such as reducing body weight and maintaining glucose homeostasis [142]. Butyrate derivative phenylalanine-butyramide restricted the damnification of reactive oxygen species to myocardium, protected mitochondrial function, and inhibited cardiac remodeling [143].

UDCA is a kind of BAs with great medicinal value for cardiovascular system in current research. It improved peripheral blood flow and liver function of patients with CHF in a RCT [144], and had antiarrhythmic and anti-atherogenic effects in animal models [145,146]. In addition, the direct activation of BA receptors is also effective. The gut-restricted FXR agonist fexaramine contributed to the remission of metabolic syndrome and reduction of systemic inflammation [147].

As for the anti-inflammatory treatment, antibiotics are the most commonly used anti-inflammatory drugs, but we do not recommend them. In addition to SCFAs and BAs we have also mentioned, there are other drugs that can reduce systemic inflammatory response. Empagliflozin—an inhibitor of sodium-glucose-cotransporter 2 always used as hypoglycemic medicine—has been approved for the treatment of HFpEF. Kolijn et al. found that an anti-inflammatory effect is also one of the pharmacological effects of empagliflozin. It could inhibit oxidative stress response and reduce myocardial inflammation through preventing the activation of PKGIαoxidation, thus protecting endothelial function and suppressing cardiomyocyte passive stiffness [148]. Anti-cytokine therapies were also proved to be preliminarily effective, for instance, IL-1 blockade with anakinra significantly reduced systemic inflammation level and improved the aerobic exercise ability of HFpEF patients [149]. Drugs targeting CCL2, IL-6, and TNF-α etc., have been proposed too, but clinical translation remains challenging [20].

### 5.7. Traditional Chinese Medicine (TCM)

TCM is a medical system based on clinical experience with a history of thousands of years. TCM herbs can modulate the composition of GMB and GBM also has a conversion effect on TCM components [150]. Several cardiovascular diseases such as hypertension, coronary heart disease, and HF can be relieved by TCM. Qishen granules, which have been widely used in ischemic CHF, adjusted the BAs profile, reversed the amino acid and fatty acid metabolism disorders and reduced the inflammation in CHF mice [151]. Moreover, some specific ingredients of TCM have been extracted to exert pharmacological effects. For example, hydroxysafflor yellow A increased the content of SCFAs in caecum by oral administration [152]; resveratrol could regulate BAs metabolism and TMAO synthesis [153]. These results indicate a potential curative effect of TCM on HFpEF. It should be noted that the complex components of Chinese herbal medicine can lead to some potential adverse effects, such as liver and kidney toxicity. Therefore, extracting active ingredients from Chinese herbal medicine is a safer and more accurate treatment and a more promising research direction.

In addition, the heterogeneity of pathophysiological mechanisms in HFpEF is an obstacle in the field of HFpEF therapy. If we could classify patients into mechanistically homogeneous subgroups, the treatment would be more precise and effective [154]. For example, patients with systemic inflammation should be treated with anti-inflammatory drugs or by targeting metabolites; dietary intervention, FMT or exercise training are applicable for patients with MetS; treatment targeting GMB may not be suitable for patients with organic heart disease such as valve diseases. How to make such a classification standard is also a complicated but significant research direction. In general, the methods mentioned above have potential therapeutic effects on HFpEF, but more studies are needed to achieve clinical transformation. Dietary therapy and exercise training can be used as supplement for existing therapies, but the treatment plan should be formulated according to the patient’s own situation.

Figure 2 shows some of the molecular pathways and potential therapeutic targets of the GMB acting on HFpEF.

## 6. Prospect

The association between GMB and various cardiovascular problems is of great concern. However, compared with atherosclerosis, hypertension, HFrEF, etc., the results about HFpEF are insufficient. The incidence of HFpEF is still increasing and the potential of therapeutic pathways targeting GMB cannot be ignored. Therefore, more studies on changes in the GMB of HFpEF are needed and the heterogeneity of GMB in different regions, races, and diets needs attention. Research on the application of FMT in HFpEF is also lacking. Drugs regulating the metabolites and inflammation deserve to be the focus due to the precision and effectiveness of their treatment. TCM has been shown to be beneficial to GMB, and finding ingredients that are effective for HFpEF is also a research direction in the future. Moreover, searching for appropriate criteria to classify HFpEF patients and achieve the goal of personalized treatment may be a practical clinical project.

## 7. Conclusions

GMB is correlated with HFpEF. The composition of GMB in HFpEF patients significantly changed, which could affect the process of HFpEF via inflammation and metabolites such as TMAO, SCFAs and BAs. Its comorbidity—MetS—also mediates the influence of GMB on HFpEF. Meanwhile, the decrease of cardiac function will undermine the intestinal barrier, result in microbial translocation then disrupt the function of the GMB. Because of these mechanisms, GMB can be used as a potential therapeutic target for HFpEF. Dietary intervention, probiotics, prebiotics, exercise training, drugs targeting metabolites and inflammation as well as TCM have been preliminarily proven to be effective, but more evidence is needed, including animal experiments and clinical studies. But it is important to note that the mechanisms in all patients with HFpEF are still unclear, so these interactions may not be consistent with all patients, which indicates to us that separating HFpEF patients into different phenotypes and looking for different treatments may be helpful.

## Figures and Tables

**Figure 1 biomedicines-11-00442-f001:**
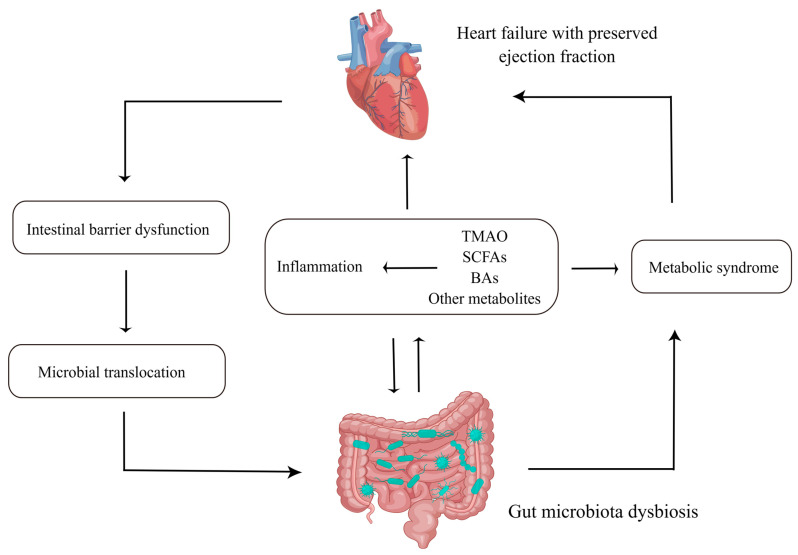
Gut microbiota (GMB) interacts with heart failure with preserved ejection fraction in multiple ways. Abbreviations: TMAO, trimethylamine N-oxide; BAs, bile acids; SCFAs, short-chain fatty acids. This figure was made by Figdraw.

**Figure 2 biomedicines-11-00442-f002:**
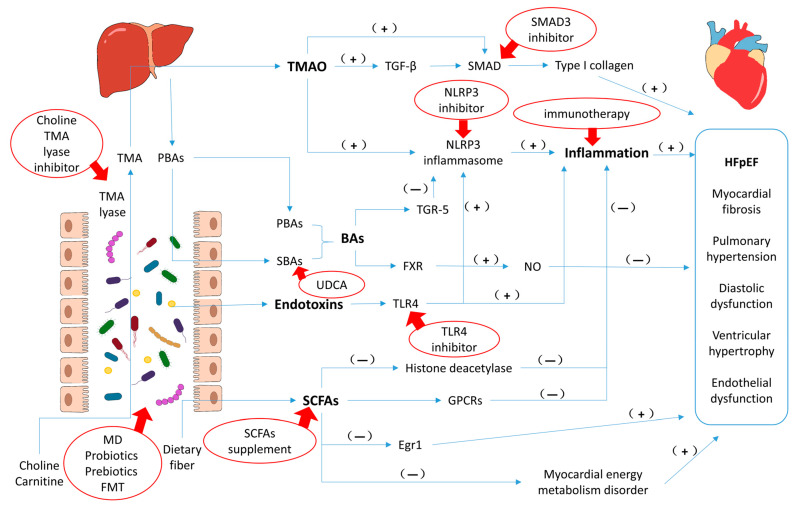
The molecular mechanisms of GMB affecting HFpEF and potential therapeutic targets. (−): Inhibit or downregulate. (+): Promote or upregulate. Red ellipses: potential therapies. Red arrows: point to the theraputic targets. Abbreviations: TMA, trimethylamine; PBAs, primary bile acids; SBAs, secondary bile acids; MD, Mediterranean diet; FMT, fecal microbiota transplantation; UDCA, ursodeoxycholic; TGF-β, transforming growth factor-β; NLRP3, NLR family pyrin domain containing 3; TGR-5, G-protein-coupled receptor 5; FXR, farnesoid X-receptor; TLR4, Toll like receptor 4; GPCRs, G protein-coupled receptors; Egr1, early growth response 1.

**Table 1 biomedicines-11-00442-t001:** Brief summary of studies investigating the features of gut microbiota (GMB) in heart failure with preserved ejection fraction (HFpEF).

Sample Size	Features of GMB Composition	Key Finding	Alpha Diversity	Beta Diversity	Reference
*n* = 30 HFpEF *n* = 30 controls	The characteristic bacterial community of HFpEF group was Enterococcus, Lactobacillus	An increase of pro-inflammatory microbiome and a decreased anti-inflammatory microbiome	Lower Chao index in HFpEF but no difference of Shannon index and Simpson index	Significant difference of composition between HFpEF and controls	[4]
*n* = 26 HFpEF *n* = 67 controls	The ratio of Firmicutes to Bacteroidetes tended to be lower (no statistical significance); significant differences in the abundance of specific bacterial populations	A depletion of the producers of short-chain fatty acids, especially Ruminococcus	Lower Chao index in HFpEF but no difference of Shannon index	Significant difference of composition between HFpEF and controls	[5]
*n* = 42 HFpEF	Firmicutes, Bacteroidetes and Proteobacteria have the highest abundance	GMB is associated with myocardial fibrosis indicators (C-terminal propeptide of procollagen type I, N-terminal propeptide of pro-collagen type III)	-	-	[7]
*n* = 42 HFpEF	The relative abundance of the most prevalent phyla was Bacteroidetes, Firmicutes and Proteobacteria	GMB is associated with the left ventricular extracellular volume	-	-	[8]
*n* = 27 HFpEF *n* = 30 age-matched controls	24 OTUs were differentially present between HFpEF and controls; the abundance of Prevotella was significantly reduced in HFpEF	The GMB differs between HFpEF and age-matched controls	-	-	[9]

## Data Availability

Not applicable.

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
