# Peer review of "The Interaction of Gut Microbiota and Heart Failure with Preserved Ejection Fraction: From Mechanism to Potential Therapies"

_biomedicines, 2023, doi:10.3390/biomedicines11020442_

Round 1

Reviewer 1 Report

1. The choice of topic for the review is probably appropriate, and there is some significance in using the metabolic syndrome as a bridge between the gut microbiome and heart failure with preserved ejection fraction, although this may be partially mentioned in a large number of other recent reviews.   2. As can be anticipated, TMAO and SCFs , BAs are reviewed in this paper, however, there is a lack of robust literature and narrative on mechanisms, after all the theme of this paper is 'from mechanism to treatment'.   3. It is also possible that the descriptions of some new metabolites have been overlooked, that one cannot stay close to the cutting edge, and that some of the most recent literature is lacking from the cited literature (most of which is before the year 2021). We know that the field of intestinal flora is changing day by day.   3. To suggest future research directions based on a synthesis of the literature is the point of the review. The article fails to provide insights under each topic heading and where insights are presented they are not sufficiently original. However, it has to be said that the presentation of "exercise" and "Chinese medicine" in terms of treatment is novel, while the FMT discussion is too brief in comparison.   4. It is disappointing that authors have made efforts to make only one figure, which also does not cover the content of the review article completely.   5. some of the literature citations are irregular in places.   6. The corresponding author appears to be a scholar in HFpEF, but there are no articles published related to gut microbiome from the group, which lacks some authority and persuasiveness.

In conclusion, the article has some innovative points, the discussion is complete, but the expression is lacking and needs a major revision.

Reviewer 2 Report

The manuscript "The Interaction of Gut Microbiota and Heart Failure with Preserved Ejection Fraction: From Mechanism to Therapy" deals with a highly interesting topic. Patients with heart failure are in a pro-inflammatory state and without doubt (in the literature),  gut microbiota have an impact on this condition. Being honest, the presented review adds no new insight on that topic, especially the focus on patients with preserved ejection fraction leaves a pretension which cannot be reached! In general the mechanisms in all patients with heart failure are still unclear, so there is no possibility to make statements about therapy. A review like "Shedding light on the possible interaction between....in (all) patients with heart failure" might be helpfull, but unfortunately the presented review has to be rejected.

Reviewer 3 Report

In this review the authors present the latest developments on gut
microbiota involvement in heart failure with preserved ejection fraction.
Accumulating evidence suggest the association between dietary and gut
microbiota dysbiosis with cardiovascular diseases mediated by several
metabolites produces by gut microorganisms. In this review the authors
focused in TMAO, SCFAs, and BAs associations with metabolic syndrome and
heart failure with preserved ejection fraction.

Round 2

Reviewer 1 Report

No further comments

Reviewer 2 Report

.